

# Virseqimprover: an integrated pipeline for viral contig error correction, extension, and annotation

Haoqiu Song[1,*], Saima Sultana Tithi[2,*], Connor Brown[3], Frank O. Aylward[4], Roderick Jensen[4] and Liqing Zhang[1]

[1] Department of Computer Science, Virginia Polytechnic Institute and State University (Virginia Tech), Blacksburg, VA, United States of America

[2] Department of Cell & Molecular Biology, St. Jude Children's Research Hospital, Memphis, TN, United States of America

[3] Department of Civil and Environmental Engineering, Virginia Polytechnic Institute and State University (Virginia Tech), Blacksburg, VA, United States of America

[4] Department of Biological Sciences, Virginia Polytechnic Institute and State University (Virginia Tech), Blacksburg, VA, United States of America

[*] These authors contributed equally to this work.

## ABSTRACT

Despite the recent surge of viral metagenomic studies, it remains a significant challenge to recover complete virus genomes from metagenomic data. The majority of viral contigs generated from de novo assembly programs are highly fragmented, presenting significant challenges to downstream analysis and inference. To address this issue, we have developed Virseqimprover, a computational pipeline that can extend assembled contigs to complete or nearly complete genomes while maintaining extension quality. Virseqimprover first examines whether there is any chimeric sequence based on read coverage, breaks the sequence into segments if there is, then extends the longest segment with uniform depth of coverage, and repeats these procedures until the sequence cannot be extended. Finally, Virseqimprover annotates the gene content of the resulting sequence. Results show that Virseqimprover has good performances on correcting and extending viral contigs to their full lengths, hence can be a useful tool to improve the completeness and minimize the assembly errors of viral contigs. Both a web server and a conda package for Virseqimprover are provided to the research community free of charge.

## INTRODUCTION

Metagenomic sequencing enables the simultaneous study of potentially billions of microbes without the need for biological cultivation. Typical metagenomic sequencing data consist of hundreds of millions of short reads (*i.e.,* computational representations of DNA sequences). Because these read lengths are short, *e.g.,* 150 bp in length, one common computational goal is to achieve longer segments *via* metagenomic short read assembly. To date, many tools such as MetaVelvet (*Namiki et al., 2012*), metaSPAdes (*Nurk et al., 2017*), Ray Meta (*Boisvert et al., 2012*), IDBA-UD (*Peng et al., 2012*), and MEGAHIT (*Li et al., 2015*) have

Corresponding author
Liqing Zhang, lqzhang@cs.vt.edu

been developed to assemble short reads into longer sequences, *i.e.,* contigs. The goal of most assembly algorithms is to generate complete genomes for the majority of organisms that are represented in the metagenomic sample (*Bickhart et al., 2022*). However, because of the complex nature of metagenomic data, for example, the presence of billions of cells, highly uneven depth of coverages of different organisms, and the presence of multiple strains of the same species, assemblers have difficulty in recovering complete genomes and often produce partial fragments of the original genomes (*Vázquez-Castellanos et al., 2014*; *García-López, Vázquez-Castellanos & Moya, 2015*; *Smits et al., 2015*). As such, it remains desirable to achieve more complete genomes from fragmented assemblies and multiple programs have been developed to extend the contigs to longer sequences and fill the gaps of draft genomes (*Boetzer et al., 2011*; *Boetzer & Pirovano, 2012*; *Farrant et al., 2015*; *Deng & Delwart, 2021*). In addition, due to the presence of closely related species and presence of multiple strains of the same species, assemblers sometimes produce chimeric sequences (sequences where genomes from multiple organisms are incorrectly assembled together *Vázquez-Castellanos et al., 2014*). Similarly, tools have been developed to correct assembly errors such as predicting the positions of chimeric sequences based on supervised learning or deep learning models (*Afiahayati, Sato & Sakakibara, 2015*; *Liang & Sakakibara, 2021*), and to improve the quality of draft assemblies by correcting single nucleotide polymorphisms, insertions, and deletions (*Walker et al., 2014*; *Wick et al., 2017*).

Active bacterial virus (phage) populations are a key example where challenges to assembly may arise (*Smits et al., 2014*; *Rose et al., 2016*; *Nayfach et al., 2021*). The presence of co-occurring strains and a high propensity for recombination and mobile element excision/integration result in fragmented or erroneous assemblies. Although currently multiple pipelines have been built to handle parts of the difficulties, there is no existing pipeline that integrates contig error correction, extension, and annotation procedures together. Here, we present Virseqimprover, an updated and integrated pipeline that allows users to perform all of these tasks in a single workflow. Specifically, Virseqimprover takes a contig and the metagenomic reads from which the contig was generated as input. Error correction and extension steps are applied iteratively to grow the contig as much as possible while ensuring that the extended contig is error free. As non-uniform depth of coverage is an indication that the assembly is likely chimeric, in the error correction step, Virseqimprover checks for the uniformity of the depth of coverage of the contig and keeps only the uniform depth of coverage part of the assembly for the next extension step. In the extension step, Virseqimprover maps reads to the edge regions of the contig (*i.e.,* left and right boundary regions), and conducts local assembly using the contig and the mapped reads. If the contig gets extended, it will be sent to the error correction step, otherwise, Virseqimprover will trim the ends of the contig and attempt to extend it further. If it is not possible, Virseqimprover will stop the error correction and extension process. Once the iterative error correction and extension steps are done, Virseqimprover will annotate the final extended contig and output the sequence along with the gene function annotation. FVE-novel (*Tithi et al., 2023*), an earlier program built by us, is a pipeline that first maps all the reads to the reference sequences using FastViromeExplorer (*Tithi et al., 2018*), performs *de novo* assembly of the mapped reads to generate contigs, and extends the contigs *via* iterative assembly to

produce final viral sequences. Unlike Virseqimprover, FVE-novel is intended to enable novel virus discovery by integrating multiple tools, including a simpler iterative assembly process. On the other hand, Virseqimprover's purpose is to recover and purify previously identified viral sequences. Specifically, we here develop an error correction and extension methodology, including uniform coverage checking, and sequence trimming during the refinement. Virseqimprover builds upon the framework of FVE-novel by extending and improving the quality of fragmented viral assemblies.

## METHODS

### Overview

Portions of this text were previously published as part of a preprint (https://doi.org/10.21203/rs.3.rs-3318217/v1). Virseqimprover was formulated for recovering complete viral genomes or viral sequences containing terminal repeats starting from single viral contigs in fasta file format and the metagenomic short reads in fastq file format. However, the steps described here could theoretically be applied to a number of difficult-to-assemble elements, including plasmids or transposable elements. The workflow can be divided into three main steps, the error correction step, the extension step, and the annotation step. The error correction step checks for both the circularity of the contig and the uniformity of the depth of coverage. The error correction and extension steps are done iteratively until the contig cannot be extended anymore. Then the final extended contig is annotated for its protein-coding regions. The output contains the extended contig along with the protein annotation. Figure 1 outlines the three steps and details are described in the following.

### Error correction

During the error correction step, the circularity of the contig is checked. Circularity is used as an indicator that the contig recovers the complete genome or it reaches to a terminal repeat, and therefore, if Virseqimprover finds that a contig is circular, it goes to the annotation step, trims the redundant part of the contig, and outputs the contig as the final contig. On the other hand, since repetitive sequences can lead to redundant local assembling process, checking circularity can avoid potential repetitive extension. Figure 2 shows how Virseqimprover checks the circularity of the contig. Assume $L_r$ is the read length and $L_s$ is the length of the contig, Virseqimprover divides the sequence into two parts, $G_a$ and $G_b$, where $G_a$ starts from $(L_s - 2 \times L_r)$ bp to $(L_s - L_r)$ bp and $G_b$ starts from the beginning or from 1 bp to the beginning of $G_a$ or to $(L_s - 2 \times L_r)$ bp. $G_a$ is aligned against $G_b$ using BLAST (*Ye, McGinnis & Madden, 2006*). If any part of $G_a$ aligns with $G_b$ with 95% identity and 95% alignment length, Virseqimprover will try to extend the alignment on both sides of the sequences to get the similar region with the maximum length. Then one of the similar regions is trimmed since having the same region twice is redundant. After checking the circularity of the contig, Virseqimprover checks the uniformity of the read coverage and uses it as an indicator for chimerism in the contig. First, per base depth of coverage of the contig is calculated using Samtools (*Li et al., 2009*). For every base position, if its coverage is within 15th to 85th percentile of all the base depth of coverages, it is considered to be within the normal range and the position is marked as normal,
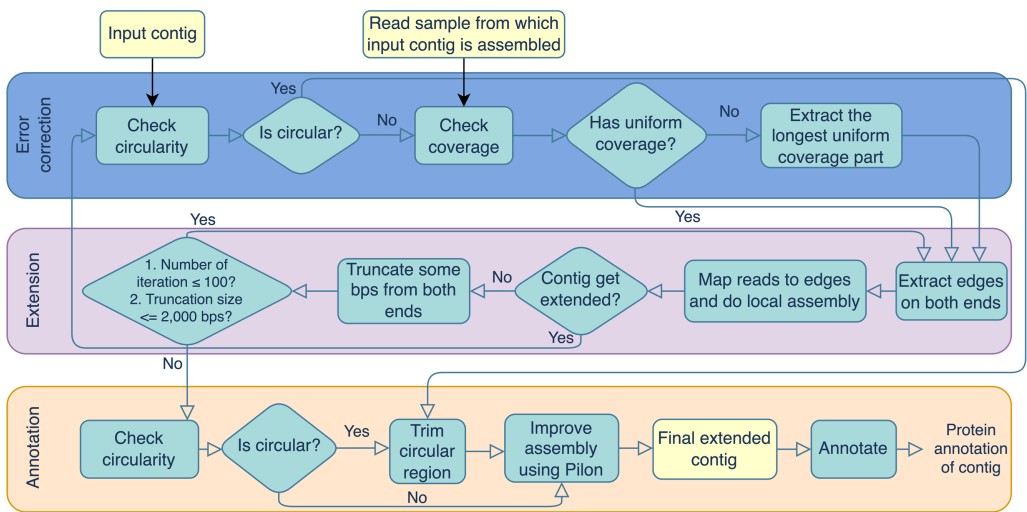

**Figure 1** Overview of the Virseqimprover pipeline, where the input is a virus contig and the metagenomic reads, the output is an extended assembly with protein annotation information.

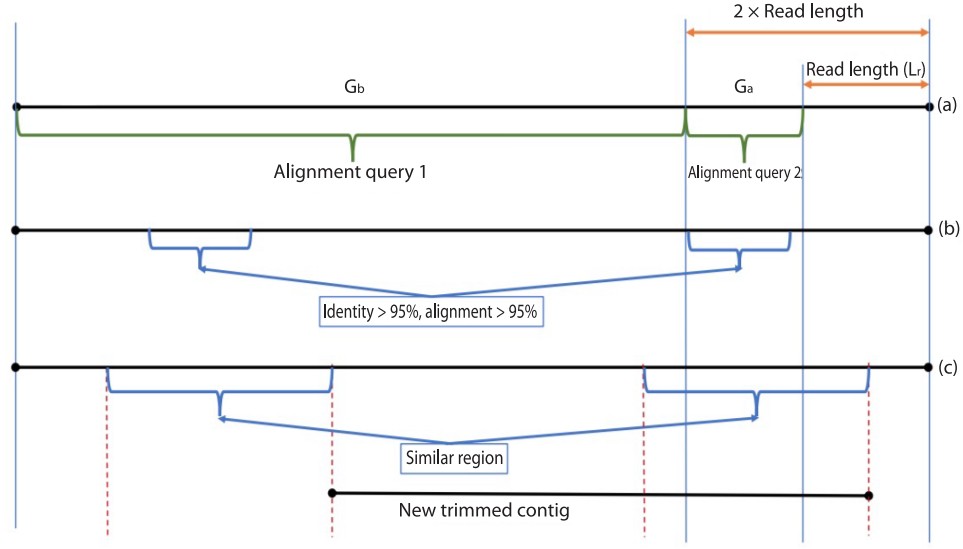

**Figure 2** Checking the circularity of the sequence.

otherwise marked as suspicious. Consecutive bases marked as suspicious form suspicious regions and those longer than 1,000 bps are considered to be true suspicious regions. All the regions other than true suspicious regions are flagged as true non-suspicious regions. Virseqimprover chooses the longest true non-suspicious region to extend during the extension step.

Virseqimprover is also applicable to linear viruses *via* its second extension termination criterion. During iterative assembly, edges are checked for circularity and, if found, trimmed

up to 2,000 base pairs and subjected to additional rounds of iterative assembly (maximum 100 iterations). If a region cannot be extended beyond this, and the longest edge is not indicative of circularity, the extension will stop and produce a linear contig, putatively representing an extended linear viral sequence.

## Extension

During the extension step, Virseqimprover first extracts the start and end edges of the contigs using BEDTools (*Quinlan & Hall, 2010*). For each edge region, read length * 1.5 is used as the default edge length. Then, for each contig, all the reads are mapped to the edges of the contig using Salmon (*Patro et al., 2017*). SPAdes (*Bankevich et al., 2012*) is used for the local assembly process. The extraction-mapping-assembly step is run iteratively for each contig until it stops growing. When the contig cannot be extended, Virseqimprover trims some bps from both ends of the contig and tries to extend the trimmed contig again. The length of the trimming part ranges from 300 bps to 2,000 bps, depending on how much trimming enables the contig extendable. The logic for trimming the ends is that our empirical investigation shows that assemblers often misassemble in one or both ends of the sequence, causing the assembler to stop prematurely which in turn leads to sequence segmentation. It is observed that trimming some bases from both ends often helps the assembler to continue the assembly in the right direction. After trimming and extending the contig, if the contig gets extended, the new extended contig goes back to the error correction step; if it cannot be extended after trimming, the extension step ends and the contig is moved to the annotation step.

## Annotation

During the annotation step, the contig is checked for circularity again. If it is circular, the contig is trimmed to remove the redundant sequence. Then Pilon (*Walker et al., 2014*) is applied to the contig to improve the assembly by correcting single nucleotide polymorphisms (SNPs), insertions and deletions. The inputs of Pilon include a genome/contig in FASTA format and reads mapped to the contig in BAM format. From the alignment information, Pilon creates a pileup structure and then corrects the base based on the frequency of each nucleotide in a position. During the base correction step, Pilon also considers if the reads are properly paired or not and the mapping quality of the base. If the alignment of read pairs indicates a discrepancy in the assembly, Pilon tries to fix the assembly by doing a local reassembly in those places. The improved contig after Pilon is sent to Prodigal (*Hyatt et al., 2010*) for ORF prediction and eggNOG-Mapper (*Huerta-Cepas et al., 2017*) for protein function annotation using the virus database. Results can be visualized using Proksee (CGView) (*Grant & Stothard, 2008*) and Clinker (*Gilchrist & Chooi, 2021*). Other than the method, other databases like Prokaryotic Virus Orthologous Groups (pVOGs) (*Grazziotin, Koonin & Kristensen, 2017*), and annotation tools like DRAM (*Shaffer et al., 2023*) and Pharokka (*Bouras et al., 2023*) can also be used to annotate the final contigs.

## DATA AND EXPERIMENTS

The input of Virseqimprover includes a contig and the metagenomic data from which the contig is generated. To generate the input contigs for Virseqimprover, we ran three assembly programs including FVE-novel (*Tithi et al., 2023*), metaSPAdes (*Nurk et al., 2017*), and MEGAHIT (*Li et al., 2015*) on two marine metagenomic samples to generate viral contigs. We then applied Virseqimprover to eight contigs generated by these tools for contig correction and extension.

### Contigs generated by FVE-novel

In this experiment, the raw assemblies was prepared by *Tithi et al. (2023)*. To be more specific, the GOV database containing 24,411 contigs as the reference "genomes" were taken and FVE-novel (*Tithi et al., 2023*) was applied to an ocean metagenomic sample (NCBI; *Sayers et al., 2022*) accession number SRX2912986 (*Aylward et al., 2017*) to generate viral contigs (*Tithi et al., 2018*). The sample contained 18,471,506 paired-end reads with an average read length 151 bp. Through FVE-novel, we chose five contigs (hereafter labeled as S0, S1, S2, S3, and S4) and applied Virseqimprover to all of them to see whether the contigs could be either further extended and/or corrected for any error. Among the five contigs, S0, S1, and S2 were highly similar to each other whereas S3 and S4 were not.

To validate the results, we reassembled the contigs using the "Map to Reference" algorithm implemented in Geneious 11.0.4 (*Kearse et al., 2012*) together with multiple rounds of manual inspection and processing. Through this semi-automated process, we hope to examine whether there are multiple viral strains or species and if there are, whether a complete assembly of the dominant strain can be generated. Specifically, the metagenome reads were aligned to contigs S0, S3 and S4 using the "Low sensitivity/Fastest" setting allowing for 10% mismatches. Then the consensus sequence from the alignment was segmented into contigs with the highest depth of coverage >40x. These contigs were binned into lists of contigs with similar depth of coverage for further assembly. Next, the contigs in each bin were iteratively grown using Geneious by mapping reads to the ends with high stringency. To be more specific, all of the phage metagenome paired-end reads were aligned to these high depth of coverage contigs using "Map to Reference" with stringent "Custom Sensitivity" settings allowing no more than 1% "Mismatches per Read" and 1% "Gaps per Read" and requiring that both of the paired-end reads map to the new consensus sequence. This process was iteratively continued until the extended contigs merged together, maintained approximately uniform depth of coverage, and could no longer be extended. Using this laborious and semi-automated approach, we recovered a 153 kb contig from contig S0, a 177 kb contig from contig S3, and a 151 kb contig from contig S4, respectively.

### Contigs generated by metaSPAdes and MEGAHIT

We also ran metaSPAdes (*Nurk et al., 2017*) and MEGAHIT (*Li et al., 2015*) on a marine plankton metagenome sample (NCBI *Sayers et al., 2022*) accession number SRX7079549 (*Beaulaurier et al., 2020*) collected from Station ALOHA in the North Pacific Subtropical Gyre to generate contigs and then applied Virseqimprover to see whether it can extend

the contigs and correct any assembly errors. These two programs (metaSPAdes and MEGAHIT) have been shown to have less misassemblies compared to some other metagenome assemblers (*e.g.*, IDBA-UD *Peng et al., 2012*) and Faucet (*Rozov et al., 2018*) as well as have good performances at the strain-level (*Wang et al., 2020*). However, many contigs generated by the two programs are highly fragmented due to uneven abundances or repeat regions (*Wang et al., 2020*). In this original study, the authors generated not only metagenomic sequencing data but also nanopore long read data which we can use to examine the performance of Virseqimprover (*Beaulaurier et al., 2020*). Based on the nanopore sequencing approach, many complete virus genome sequences were recovered. We BLASTed the contigs generated by metaSPAdes and MEGAHIT against the two recovered complete virus genomes AFVG_25M466 and AFVG_25M409, and identified three contigs that are highly similar to the genome sequences. The three contigs, referred to as S5, S6, and S7, were sent to Virseqimprover for further extension.

### Evaluation of virseqimprover

A simulation experiment was conducted to evaluate the capability of correction and extension of Virseqimprover. We injected the genome of a papillomavirus (genome accession GCF_002826985.1) into the middle of contig S7 as a misassembly, generated 2,000 number of reads targeting 8X depth, spiked the reads into the original metagenomic sample that we used to generate S7 (SRX7079549), and ran MEGAHIT to assemble the spike-in sample. We identified the viral contig from the assemblies and ran Virseqimprover to evaluate whether it could correct remove the misassembly and extend it further.

Moreover, with regards to the state-of-the-art contig extension tool, ContigExtender is a popular tool that is used to extend assembled contigs following *de novo* assembly. ContigExtender uses a recursive overlap layout candidates strategy that explores multiple extending paths to achieve longer contigs (*Deng & Delwart, 2021*). Here we ran ContigExtender and compared the extended contigs with those extended by Virseqimprover using CheckV (*Nayfach et al., 2021*) for contig completeness and quality. CheckV determines the completeness and quality of assembled contigs by comparing them to IMG/VR (*Camargo et al., 2023*) database, which is a large database of complete virus genomes and has been used widely to evaluate the quality of the assembly.

## RESULTS

### Contigs S0, S1, S2

Among S0, S1, and S2, S0 is the longest with length 193,112 bp. Figure 3A shows that for S0, depth of coverage from around 24,500 bp to 25,500 bp is lower than the average depth of coverage and varies a lot after 150 kb. Virseqimprover checked for the uniformity of the depth of coverage and extracted the longest region with uniform depth of coverage, which was a region with length 127,423 bp from 25,567 bp to 152,989 bp. This longest non-suspicious region went through the iterative extension and error correction steps. When the iterative extension step was done, the output contig (length 163,662 bp) was checked for circularity. When the circular region was trimmed and Pilon was applied to the output contig, an improved contig (length 152,707 bp, denoted as S0′) was generated
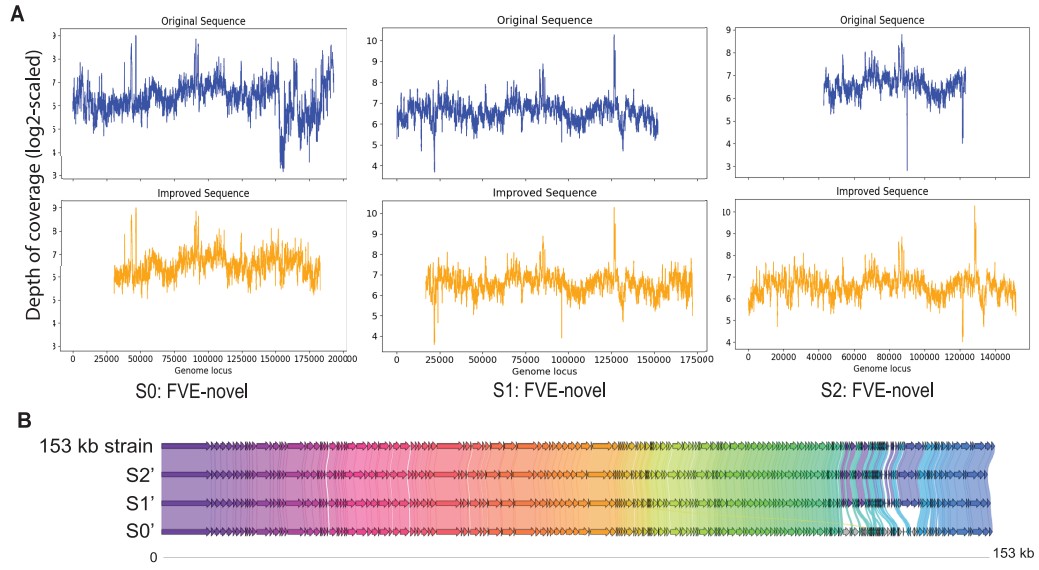

**Figure 3** (A) **Depth of coverage of the original and improved contigs S0, S1, and S2. (B) Visualization of gene cluster comparison of the 153 kb strain with the improved contigs S0, S1, and S2.**

and became the final output of Virseqimprover as shown in Fig. 3A. Some missing part on the left was due to the trimming of the circular region.

Similarly, examination of the depth of coverage along S1 revealed a non-uniform depth of coverage region or suspicious region from 133,376 bp to 134,543 bp. Thus, Virseqimprover then extracted the longest non-suspicious region which was a region of length 133,375 bp from 1 bp to 133,375 bp and then applied the iterative extension and error correction steps to generate an extended contig. When all these steps were finished, the circularity of this contig was checked and Pilon was applied to the contig. The final output was a contig with length 152,362 bp. Figure 3A shows the depth of coverage along both original (S1) and final sequences (S1′).

For contig S2 with length 80,620 bp, the per base depth of coverage was checked and Virseqimprover identified no non-uniform depth of coverage region or suspicious region. Hence this contig directly went through the iterative extension and error correction steps. Finally, after using Pilon to improve the assembly, a contig of 151,828 bp was generated. Figure 3A shows that after applying Virseqimprover, we extended the original contig on both ends and nearly doubled the total length of the original contig to get a greatly extended contig (S2′) which has a uniform depth of coverage along the sequence.

The improved versions of contigs, S0′, S1′, and S2′, were compared to the 153 kb strain of a novel uncultured virus. After predicting the genes and visualizing the gene cluster comparison, as shown in Fig. 3B, we can see that S1′ and S2′ are very similar to the 153 kb reference strain, whereas S0′ is a bit different from all other contigs. This shows that Virseqimprover has correctly recovered the whole virus sequences. Additionally, since some regions in S0′ do not match with any of the regions of all the other contigs, it could be an indication that S0′ is a different strain of the same virus species.
## Contig S3

In the same manner, S3, with length 132,604 bp was checked for the uniformity of the depth of coverage and found that it had a region with low depth of coverage at around 49 kb to 66 kb as shown in Fig. 4A. BLAST search shows that this region aligns best with Cyanophage P-RSM3 and Prochlorococcus phage P-SSM4 whereas the other parts of the contig aligns best with Cyanophage P-RSM1 and Synechococcus phage metaG-MbCM1, indicating that this low depth of coverage region is probably a misassembly. Virseqimprover flagged all the suspicious regions and extracted the longest region with uniform depth of coverage from 66,390 bp to 132,603 bp. Then this 66 kb region was extended through iterative extension and error correction and a contig with length 160,821 bp was generated. After checking for the circularity of this contig and applying Pilon to this contig, an improved contig (denoted as S3′) with length 160,744 bp was produced which was the final output of Virseqimprover. Figure 4A shows that the non-uniform regions in the original contig are filtered out so that the final corrected and extended contig has a uniform depth of coverage along its length. Using BLAST search, we identified four phages that show the highest sequence similarity to S3′. Based on the gene cluster comparisons, as shown in Fig. 4B, we can see that the improved contig S3′ does have some similar proteins with these phages. However, DNA sequence alignment of S3′ with these genomes also reveals some dissimilar regions, with great sequence identity variation along the entire sequence, ranging from 75.38% to 82.72%. Hence the improved S3′ might be from a novel phage species.

Figure 4C shows the similarity of S3′ (length 160,744 bp) to the viral sequence (177,631 bp) recovered by the semi-automated assembly process in Geneious. The protein identity threshold is 30%, which means that two proteins are considered to belong to the same group if their protein identity value is above 30%. Apart from the beginning part in the 177 kb strain that does not have many alignments in S3′, a small region in the middle of the sequence also shows difference between these two sequences (colored as the gray arrows). We thus further compared S3′ with the 177 kb strain to find out the difference in this specific region. Based on the pairwise DNA sequence alignment by EMBOSS Stretcher (*Madeira et al., 2022*), the comparison of these two contigs reveals that in the 177 kb strain, a 1,282 bp region from 56,831 bp to 58,112 bp does not have many matches with S3′. In this part instead of this 1,282 bp region, S3′ contains a 1,342 bp region from 22,163 bp to 23,504 bp. Analysis of the depth of coverage of these two sequences in the area where they are different reveals that those areas have a relatively lower depth of coverage (about 150x) compared to the average depth of coverage (about 300x). Moreover, based on the BLASTP search, the specific protein sequence corresponding to this area in contig S3′ aligns best with Synechococcus phage metaG-MbCM1, whereas the 177 kb strain aligns better with Synechococcus phage S-SM2. The differences between S3′ and the 177 kb strain suggest that they may represent different strains of the same phage.

## Contig S4

S4 has 136,254 bps. After Virseqimprover's contig extension, error correction, and circularity check, S4 was extended to 151,190 bps. Figure 5 shows that the depth of coverage of the original S4 is rather uniform, and was extended for both sides of the

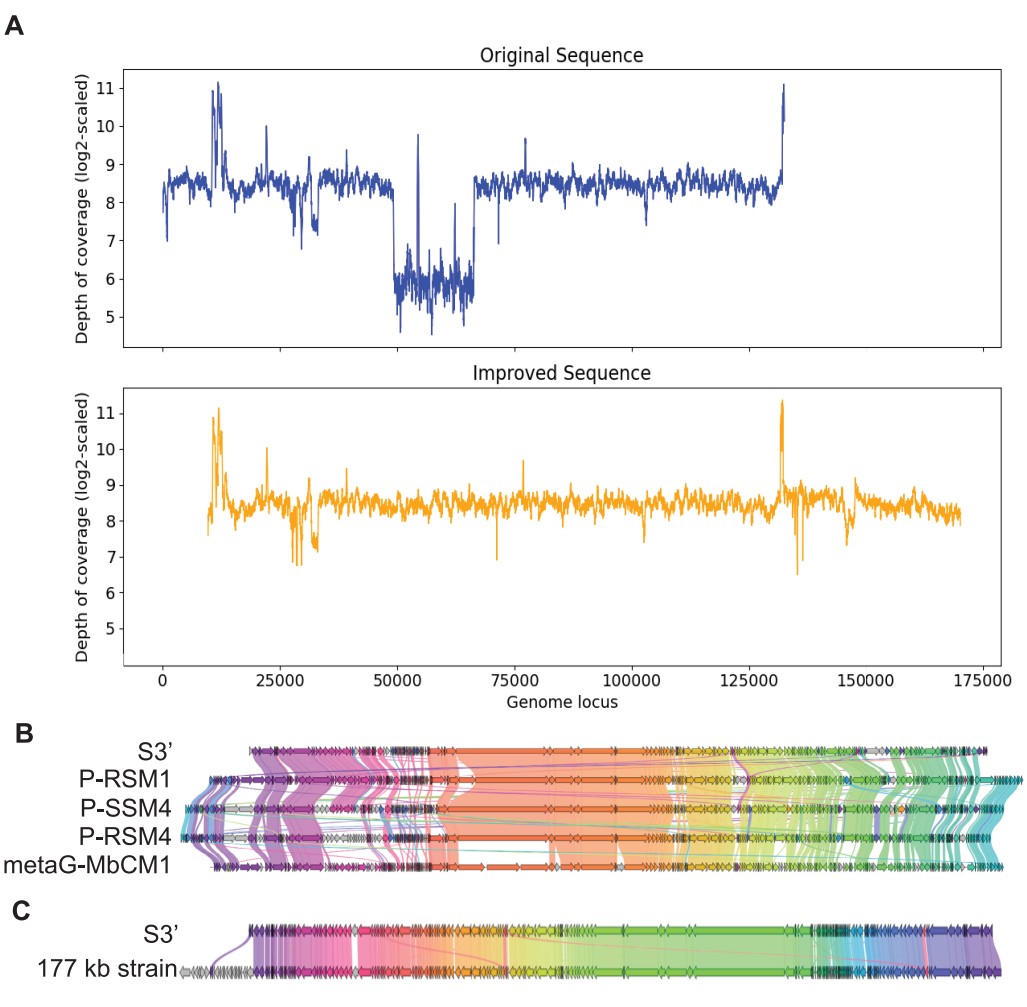

**Figure 4** **Visualizations of depth of coverage and gene cluster comparisons of contig S3.** (A) Depth of coverage of the original and improved contig S3. (B) Visualization of gene cluster comparison between improved S3 and similar strains. (C) Visualization of gene cluster comparison between improved S3 and the reference 177 kb strain.

sequence, with both left and right ends of S4′ showing higher depth of coverage than nearby regions, which indicates the presence of repeats. Closer examination of the end sequences reveals that the regions indeed are repeats.

## Contig S5, S6

S5 is generated by MEGAHIT and contig S6 is generated by metaSPAdes. These two contigs both have a 99% identity to the marine virus with ID AFVG_25M466, covering 12% and 40% of the viral genome, respectively. After applying Virseqimprover, S5 got extended from 4,179 bp to 14,374 bp, and S6 extended from 13,396 bp to 22,526 bp (Fig. 6). The extended S5′ covers 43% of the marine virus genome, and the extended S6′ covers 68%. Pairwise alignment between the extended sequences and the marine virus genome shows that the extended parts are identical to the corresponding parts of the marine virus genome,

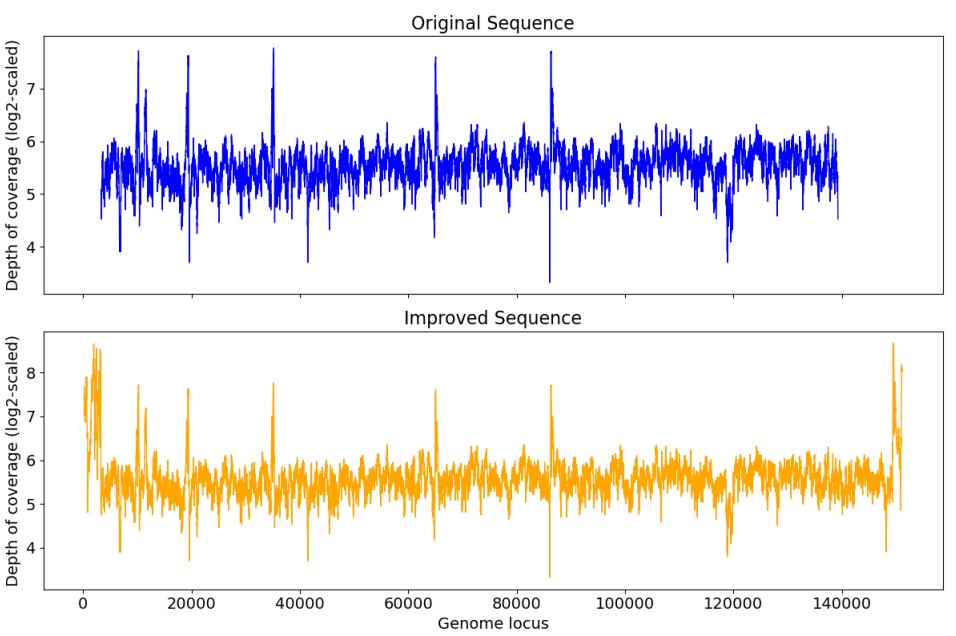

**Figure 5** Depth of coverage of the original and improved contig S4.

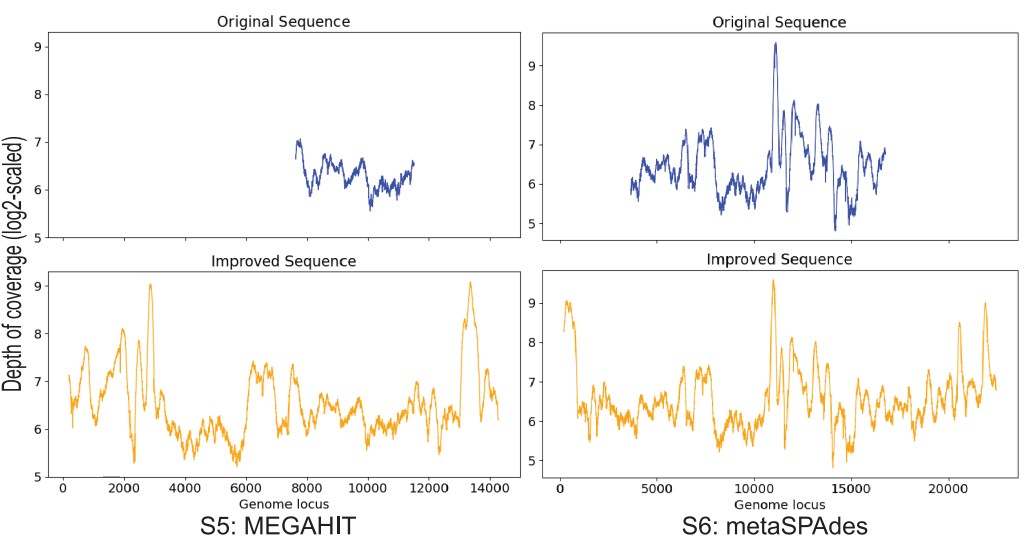

**Figure 6** Depth of coverage of the original and improved contigs S5 and S6.

suggesting that Virseqimprover can accurately extend the contig sequences generated by other assemblers.

## Contig S7

S7 is generated by metaSPAdes. It has a 99% identity to the marine virus with ID AFVG_25M409. After applying Virseqimprover, S7 was extended from 23,114 bp to

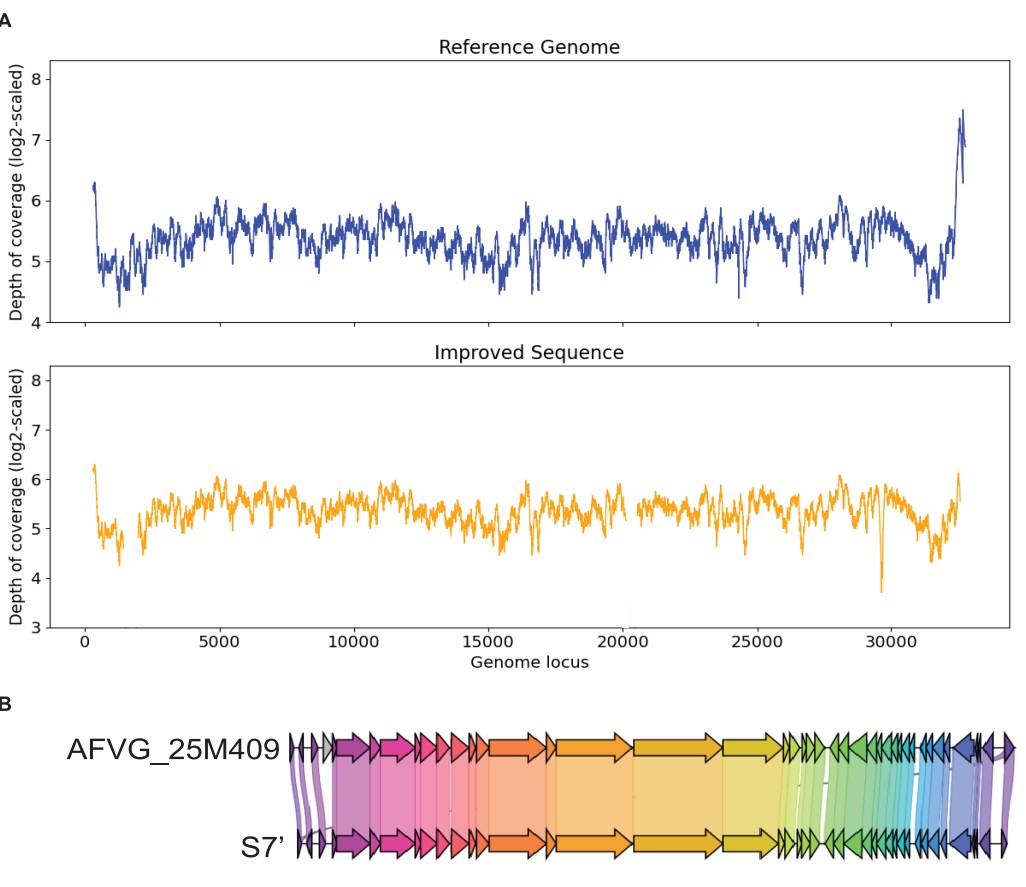

**Figure 7** (A) Depth of coverage of the reference genome and improved contig S7. (B) Visualization of gene cluster comparison between the reference genome and the improved contig S7.

32,035 bp, which covers 98% of the AFVG_25M409 viral genome (32,812 bp). Figure 7 shows that Virseqimprover successfully extends the original fragmented contig to nearly complete genome with high accuracy.

## Simulation experiment evaluation

In order to examine the capability of Virseqimprover to remove misassembly within sequences, a papillomavirus genome was injected into the sequence of original S7 at a random position. After generating and spiking in the 2,000 synthetic reads and running MEGAHIT to assemble the spike-in sample, a contig with length 31,745 was found to have 99.97% identity with Marine virus AFVG_25M409 and 100.00% identity with the papillomavirus, respectively. Then we ran Virseqimprover on the contig with the insertion and got a sequence with length 32,395 bp. This revealed that the injected genome was entirely removed and the pure AFVG_25M409 viral genome was fully recovered. Figure 8 shows the depth of coverages of the original and recovered sequences, and the low depth of coverage region within the original sequence corresponds to the injected viral genome.
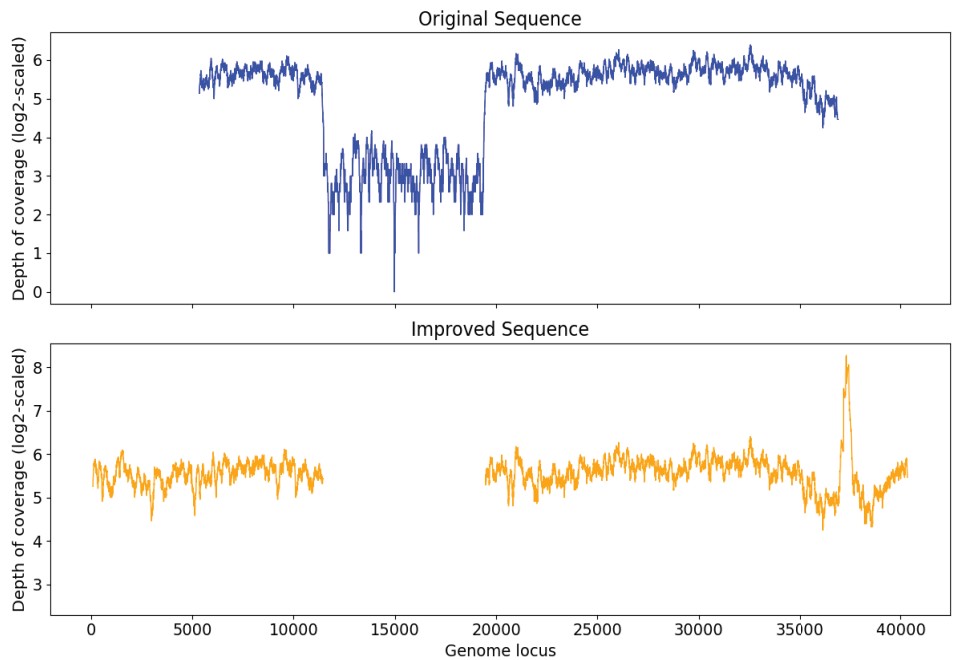

**Figure 8** Depth of coverage of the original injected sequence and the recovered sequence by Virseqimprover.

## Contig completeness and quality evaluation

Figure 9 shows the completeness of both the original and refined sequences by ContigExtender and Virseqimprover. Based on CheckV, only one of the eight contigs generated by either FVE-novel, metaSPAdes or MEGAHIT is complete. In contrast, four contigs recovered by Virseqimprover are complete, thus achieving 37.5% improvement over these assembly tools in contig completeness. ContigExtender did not generate any complete contigs (S0 is already 100% complete and ContigExtender extended another 256 bp). Remarkably, after applying Virseqimprover, the completeness of S2 improved from 48.19% to 100%, S4 from 60.72% to 100%, and S7 from 68.89% to 100%. S5 also shows a significant improvement of completeness from 13.18% to 44.92%, compared to 15.16% by ContigExtender. This shows that Virseqimprover is effective in extending contigs and improving the completeness of contigs generated by other assemblers or contig extension tools.

Table 1 shows the sequence quality assessed by CheckV as well as length information for all the contigs generated by the original assemblers and extended by either ContigExtender or Virseqimprover. As for the quality, CheckV has four categories of increasing quality: low, medium, high, and complete. Virseqimprover improved the quality of six of the eight contigs, compared to only one by ContigExtender. Moreover, four of the eight contigs extended/refined by Virseqimprover are complete, two high, one medium, one low, compared to two high, three medium, and three low for the original contigs, and two high, four medium, and two low by ContigExtender. Thus, Virseqimprover produces
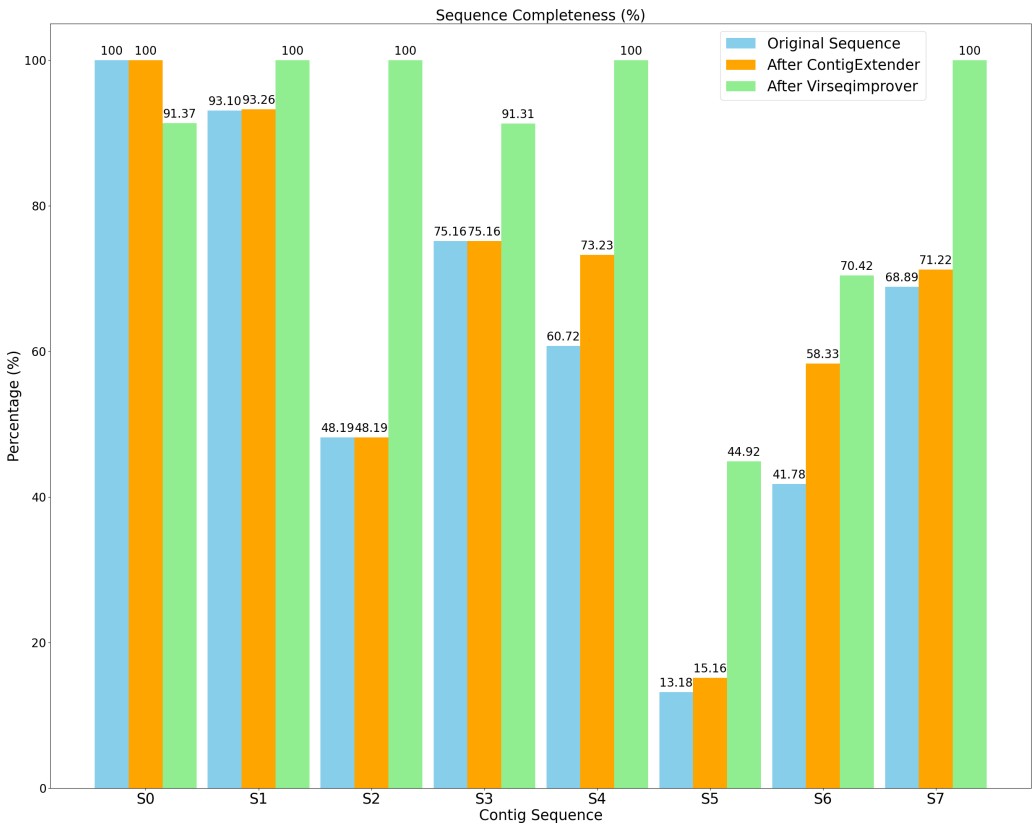

**Figure 9** Sequence completeness of original contigs, contigs after ContigExtender, and contigs after Virseqimprover.

better quality and more complete sequences than the several commonly used assemblers as well as ContigExtender, a tool designed specifically to extend contigs. Interestingly, due to the nonuniform depth of coverage, Virseqimprover trimmed part of S0 that has low depth of coverage, which made the final sequence shorter. Taking together, Virseqimprover significantly enhances the quality of the contigs and is able to extend fragmented sequences into more complete sequences or full genomes. As ContigExtender and Virseqimprover are both contig extension tools, it warrants a detailed comparison between the two.

ContigExtender focuses on consensus read mapping and does not consider uniform depth of coverage and early stopping conditions. Also, ContigExtender runs only on Docker images, so it is not as flexible as Virseqimprover especially when users on HPCs do not have admin permissions. Instead, Virseqimprover not only provides a user-friendly web server but also a Conda package so that all users can run the tool easily on any platform. Another limitation of ContigExtender is that since the input contig is kept entirely for extension, if there is any misassembly within the sequence, ContigExtender will not correct it and extending the misassembled sequence longer only leads to longer misassembled sequence. In contrast, Virseqimprover considers error-correction and extension together when running the iterative local assembly process, and also trims some bases in the sequence

**Table 1 Comparison of lengths and qualities of contig sequences.**

| Contig | Original sequence | | Sequence from ContigExtender | | Sequence from Virseqimprover | |
|---|---|---|---|---|---|---|
| | Length (bp) | Quality | Length (bp) | Quality | Length (bp) | Quality |
| S0 | 193,112 | High | 193,368 | High | 152,707 | High |
| S1 | 155,659 | High | 155,912 | High | 152,362 | Complete |
| S2 | 80,620 | Low | 80,620 | Low | 151,828 | Complete |
| S3 | 132,604 | Medium | 132,604 | Medium | 160,744 | High |
| S4 | 136,254 | Medium | 136,510 | Medium | 151,190 | Complete |
| S5 | 4,179 | Low | 4,807 | Low | 14,374 | Low |
| S6 | 13,396 | Low | 18,664 | Medium | 22,526 | Medium |
| S7 | 23,114 | Medium | 23,928 | Medium | 32,035 | Complete |

during the iterative assembly process to avoid early stopping when the extension gets terminated. As a result, Virseqimprover will conveniently get longer, more complete and more accurate sequences for the input sequences.

## DISCUSSION

In this article, we developed Virseqimprover, a computational pipeline for improving the completeness of viral assemblies. Virseqimprover leverages several enhancements over available methodologies, including the splitting of chimeric sequences based on uniformity of depth of coverage, and edge-trimming prior to extension of viral contig length *via* local assembly. Virseqimprover was developed to improve assembly of viruses, however, aspects of the pipeline are theoretically applicable to any target.

Our results indicate that Virseqimprover successfully extended and corrected errors for all the contigs produced by a range of different assembly programs. For each contig, the alignments of the original and final improved contigs which shows the identity and mismatch of these original and final sequences are shown (Supplementary Material). Comparison of the extended contigs with the known reference strains shows that the extended contigs have high similarity to them, suggesting that our tool successfully corrected and extended those contigs to as close to their full lengths as possible. As a result, due to the fact that it is challenging for current assemblers to produce complete virus genomes from metagenomic data, provides surely become a useful tool to help the assemblers to generate the viral contigs correctly to nearly their full lengths. The runtime and hardware usage are heavily dependent on the input contigs and samples. For example, S7 was run with a 20 GB metagenomic sample to get the final improved sequence for a total running time of 2.91 h, and max memory usage of 1.14 GB. Larger samples and larger number of iterative assembly processes will cost more time to run the whole pipeline.

Despite the advantages of Virseqimprover on correcting and extending viral contigs from metagenomic reads, Virseqimprover also has some limitations. One limitation is that during the depth of coverage checking step, Virseqimprover does not check the GC content of the suspicious regions. But in Illumina sequencing, very high or very low GC

content (>70% or <30%) can result in reduced mapping depth of coverage and higher error rates. As a result, a low depth of coverage region with high or low GC content can be actually part of the contig, while Virseqimprover can wrongly mark it as a suspicious region and discard that region. Another limitation is that it can incorrectly mark a linear phage as a circular one. Some linear phages may have repetitive sequences at the ends. Because of these repetitive sequences, assemblers can start the assembly of the phage again from the beginning (*Garneau et al., 2017*; *Boeckman et al., 2024*). During the circularity checking step, Virseqimprover might incorrectly mark this phage as a circular genome. Future work is needed to address this limitation. Besides, Virseqimprover is not suitable to extend every contig in the metagenomic samples where the amount is huge. Instead, the main significance of the tool is to correct and extend a small amount of contigs that researchers are particularly interested in. Moreover, since some of the sequencing data (*e.g.*, amplicon enrichment sequencing data) can generate significant variation in site-wise depths of coverages, considering more features like paired-end information, haplotype block data, read mapping quality, and the distribution of read ends when recovering those non-uniform sequencing depths and identify incorrect misassemblies more precisely is valuable for future exploration.

## ACKNOWLEDGEMENTS

The authors would like to acknowledge Muhit Islam Emon (Virginia Tech) for help on the web server deployment.

### Funding

This work was supported by funds from the National Science Foundation (NSF: # 2004751, # 2125798), and by funds from Centers for Disease Control and Prevention (CDC BAA: 75D301-22-R-72097). There was no additional external funding received for this study. The funders had no role in study design, data collection and analysis, decision to publish, or preparation of the manuscript.

### Grant Disclosures

The following grant information was disclosed by the authors:
The National Science Foundation: NSF: # 2004751, # 2125798.
Centers for Disease Control and Prevention: CDC BAA: 75D301-22-R-72097.

### Competing Interests

The authors declare there are no competing interests.

### Author Contributions

- Haoqiu Song conceived and designed the experiments, performed the experiments, analyzed the data, prepared figures and/or tables, authored or reviewed drafts of the article, and approved the final draft.

- Saima Sultana Tithi conceived and designed the experiments, performed the experiments, analyzed the data, prepared figures and/or tables, authored or reviewed drafts of the article, and approved the final draft.
- Connor Brown conceived and designed the experiments, authored or reviewed drafts of the article, and approved the final draft.
- Frank O. Aylward conceived and designed the experiments, authored or reviewed drafts of the article, and approved the final draft.
- Roderick Jensen conceived and designed the experiments, authored or reviewed drafts of the article, and approved the final draft.
- Liqing Zhang conceived and designed the experiments, authored or reviewed drafts of the article, and approved the final draft.

## Data Availability

All data and codes are available at GitHub and Zenodo:

- https://github.com/haoqiusong/Virseqimprover.
- Song, H. (2024). Virseqimprover. Zenodo. https://doi.org/10.5281/zenodo.13743177
The sequences are available at NCBI: SRX2912986, SRX7079549.
https://www.ncbi.nlm.nih.gov/sra/SRX2912986
https://www.ncbi.nlm.nih.gov/sra/SRX7079549

## Supplemental Information

Supplemental information for this article can be found online at http://dx.doi.org/10.7717/peerj.18515#supplemental-information.

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
