# Peer review of "Virseqimprover: an integrated pipeline for viral contig error correction, extension, and annotation"

_PeerJ, doi:10.7717/peerj.18515_

## Round 0.1 · original submission · Major Revisions

Based on three reviews of your manuscript, a major revision is required before we will be able to determine if the manuscript can be accepted for publication. Your revision will need to address all of the numerous concerns raised in the reviews. And while all of these individual concerns need to be addressed, please pay particular attention to the following:

1. To what extent does Virseqimprover require an accurate annotation of the contigs and reads as coming from virus genomes? And if this is there case, does this requirement impose a significant limitation to program functionality?

2. Please address the program limitations with regards to accurate assembly of linear virus genomes.

3. As indicated in all three reviews, the figures will need significant revision and possibly consolidation.

4. Please be sure that all raw data used to support this work is available in GitHub or another public repository.

·

Basic reporting

- The use of the English language is accurate and professional and therefore I have no remarks on this issue.
- In the introduction, the list of programs in lines 38-39 is not chronologically sorted and this is confusing. In addition, I would have explicitly used the name of the programs before the literature citations. For example:
"Consequently, programs such as sspace Boetzer et al. (2011); Boetzer and gapfiller Pirovano (2012); Wisescaf-folder Farrant et al. (2015); Contigextender Deng and Delwart (2021) have been developed..."
-In general, there is no chronological order (or any other logical one) when citing the list of references in the text. See also lines 30-31, and lines 44-45. Is there any reason for this? If not, I would suggest being more systematic when arranging bibliographic citations.
-Another issue I find is the fact that authors do not indicate unambiguously in the introductory section the fact that the primary target of their pipeline are the viral metagenomes. They state the metagenomic sequencing data in the beginning of their introduction, but it remains unclear when they refer specifically to viruses, as those entities and their associated problems are not explicitly mentioned in the introduction. For this reason, I have not clear whether the difficulties e.g. chimeric sequences, assembly errors, etc. enumerated in the first paragraph are referred to any metagenomic data or are exclusive or particularly exacerbated in the case of viruses.
- Although not necessarily critical for their purpose, most of the figures have a resolution that could be improved for final stages, because their associated text is sometimes barely readable.
- I am not fully convinced regarding the number and distribution of the figures. The list of figures and the way they appear can be misleading. Is there any reason to keep figures 3, 5 and 7 as separated ones, when they show the same type of result? This distribution leads to forced leaps to the reader such as having the results for contig S7 split between two figures and in reverse order, first figure 7(3) and later figure 4(4), to keep the same order as other contigs. In addition, there is no mention to Figure 6 in the text. Therefore, I would recommend grouping Figures 3, 5, and 7 into a single paneled figure, and properly citing Figure 6, or otherwise deleting it. And structure the text based on this arrangement.

Experimental design

- Regarding the methodology, my main concern is the fact that the authors state that the input of Virseqimprover is a viral contig (and the metagenomic short reads from which the input contig was assembled), which implies that it has to be previously identified and annotated as such. In actual metagenomic samples many of them very complex in their composition, this prior identification is the main challenge to be addressed, as many samples contain many non-viral reads, actually those non-viral reads can constitute the majority of the reads. How do the authors deal with this issue? Because, having previously identified viral contigs, the author's pipeline is able to extend the length of those contigs, which is ok, but it represents "only" a minor improvement compared to the major limitations of identifying which input contigs will be processed through their pipeline because they are undoubtedly viral. I miss an explanation regarding how the authors deal with this crucial issue and make a decision regarding which contigs are eligible and which ones are not for their pipeline.
In another way to formulate my concern/question related to this, can indeed this pipeline be used to extend any contig, regardless of its viral or non-viral nature?

-Another concern is regarding the circularity. Probably I missed something, but, in the case of linear genomes, how can be circularity be used as an indicator of complete genome recovery? Do you mean a kind of overlapping of similarity of the edges beyond the linear genome molecule ends? In this regard, figure 2 (whose resolution could be improved) is relatively clear compared to the explanation (lines 76-79) which is cumbersome, and it is not easily understood why the authors chose those parameters.

-Regarding the chimericness in the contig, not a criticism but rather a curiosity, is there any reason to justify or support the percentiles 15-85 chosen by the authors?

- I find the extension step as the main strength of the Virseqimprover pipeline, in contrast to my criticism to the previous two steps. In this case, authors support their justification on empirical investigations and try to overcome the premature stop. However it is not fully clear why (lines 101-102) "trimming some bases from both ends often helps the assembler to continue the assembly in the right direction". Do the authors have a clue about how this observation can be explained?

-Some additional information regarding the datasets used (e.g. type of sample, etc.) would have been useful. Also, which are the reasons underlying the use of each one of the three selected assemblers (FVE-novel, metaSPAdes and MEGAHIT) on a different dataset or, when used on the same dataset (metaSPAdes and MEGAHIT on SRX7079549) of generating different contigs instead of comparing the same contig but generated with alternative assemblers? Probably it is justified to encompass different assemblers and different contigs to compare with Virseqimprover, but as far as I know it is not clearly explained in the text.

Validity of the findings

-The findings presented in this work are potentially valuable for a particular objective of extending assembled contigs to complete or improve annotation of viral genomes while maintaining extension quality. The authors show this improvement through a series of chosen examples, and the discussion and conclusions are quite clear and convincing. In particular, the comparison with ContigExtender is interesting and one of the strengths of this paper. However, I am convinced that if the way in which the results are presented were more precise and better structured, the message would be transmitted in a better way.

Reviewer 2 ·

Basic reporting

The article is generally well written and the points made clear, with appropriate background and discussion material. Further references were sufficient and appropriate with one notable exception: NCBI accessions are reference multiple times, including with explicit reference to "NCBI" but without any citation for the database.

The figures are of generally high quality; however, I have a few suggestions for improvement. I believe figures 4, 5 and 7 would be greatly improved if some sort of "zoomed-out" alignment of the contigs before and after application of the pipeline being presented, color coded based on identity/mismatch, was included with the graphs. Coverage information is important and relevant, but ultimately what is obtained is a consensus sequence, and seeing changes across the length of the contig would be very beneficial. While the regions that look consistent before and after the processing may very be identical, the refinement process may have changed some individual bases at the consensus level and this should be indicated. Figure 4.4's placement is slightly odd to me. I understand that its the same style of figure as others in 4, but the information is more closely related to that in Figure 7. In figure 8, it would be useful to indicate, for each bar, what reference was used by CheckV. If the same reference was used for each bar in a single group of three, than perhaps it'd be sufficient to note this in the results section.

More generally, I have 3 main issues I believe should be addressed prior to publication. 1) At the end of the results section I believe dockerization is grossly mischaracterized. While many people do indeed prefer conda or pip type install patterns, one of the main motivations of docker is to simplify installation of dependencies, ensure reproducibility via packaging dependencies in a specific build/version with the software in the container and to avoid issues with trying to execute software on a variety of OS's. Further, docker is widely used and greatly simplifies deployment of code on clusters in HPC and cloud environments. I do not think it detracts from their software for it not to be containerized, but to state that a conda install pattern is more flexible than a docker image is inaccurate. 2) One component of the pipeline the authors are trying to simplify includes annotation of contigs, but this component of the write-up is exceptionally weak. I believe comparison of annotation results to other annotation pipelines similar to what was done for other steps of the workflow should be included so that all aspects of the pipeline can be supported with a similar depth of analysis. 3) Given they are presenting a software tool, and that the quality of the results is not the only factor that influences choice of software package, I think it would be good to include information on runtime performance metrics, such as CPU time, wall clock time, max memory usage, disk space required for reference data, etc.

Experimental design

While I think the manuscript was generally well motivated and investigated, I think for an audience not necessarily as familiar with this topic, it may be worth clarifying why the specific investigations were conducted. For example, if someone were two write their own workflow with commonly available tools, they may not necessarily see the improvements in contig quality presented here for lack of trimming during the refinement process before trying to recruit additional reads. More explicit statement of the types of problems that are being addressed with these investigations, that make the work more than just a convenient pre-written workflow, would make these points more likely to be appreciated by a wider audience.

I previously noted concerns about the investigation into the annotation component and performance metrics.

Validity of the findings

While I find the results presented clearly and the conclusions well supported in general, I was not able to locate the raw data presented here (contigs, tabular results present in figure like figure 8, etc.) Including these in a directory in the github e.g. would be sufficient IMO.

Additional comments

While its noted on the github page, I think mention of input and output file formats in the methods section would be appropriate.

Reviewer 3 ·

Basic reporting

The manuscript contains incorrect use of tenses. Here are specific examples from one paragraph:
• Lines 129-130: “The sample contains 18,471,506 paired-end reads ….” should be revised to “The sample contained …”
• Lines 133-135: “We applied Virseqimprover to the longest five contigs … to see whether the contigs can be either further extended and/or corrected for any error.” should be revised to “… to see whether the contigs could be …”
• Lines 135-136: “Among the five contigs, S0, S1, and S2 are highly similar to each other whereas S3 and S4 are not.” should be revised to “Among the five contigs, S0, S1, and S2 were highly similar to each other whereas S3 and S4 were not.”

The manuscript lacks sufficient detail in certain sections, making it difficult for readers to understand the context and significance of the work. Specific examples include:
• Lines 37-40 and 42-46: The names of the programs should be explicitly mentioned instead of only the names of the developers.
• The functionalities of currently available programs should be reviewed and compared more thoroughly to provide a comprehensive overview of the field and current existing gaps.
• The datasets SRX2912986 and SRX7079549 have been previously analysed. However, the manuscript provides a minimal overview of the data without clarifying what should be expected in a more wholistic manner.

Regarding the analysis of SRX2912986 described under the section “Contigs generated by FVE-novel” (which, based on the text, should be renamed to “Contigs generated by FVE-novel and Geneious”), I found that the text largely describes previously published work by this group of authors (PMID: 36607772; I cannot retrieve the full paper, but I found a preprint on researchgate), The authors should be clearer what had previously been done, and what was done in addition in this work.

Many figures contain illegible text. The order of the figures is incorrect, e.g. Figure 9 is mentioned before Figure 6-8 in the text. In addition, the way figures are labelled are sometimes inconsistent. For example, in Figure 4, some contigs have their own explicit descriptions next to them, while others are labelled with just letters, with descriptions provided in the figure legend.

The names of the contigs (“S#”) are not informative and are difficult to track across the manuscript.

The terms “depth”, “depth of coverage” and “coverage” are used interchangeably throughout the text. In a broader context, “coverage” can also refer to “the percentage of the genome covered by sequencing”, which can be confusing.

In addition, the documentation for the program on both the web server (http://virchecker.cs.vt.edu/virseqimprover) and GitHub (https://github.com/haoqiusong/Virseqimprover) are still underdeveloped.

Experimental design

There are several issues with how the program works.

The authors use contig circularity as an indicator of genome completeness (lines 71-72). While this is reasonable for bacterial genomes, it is well known that not all viruses have circular genomes, so I do not see how this is generally applicable with viruses. The authors should consider other additional criteria to evaluate genome assembly completeness.

If a contig has similar sequences at its two ends, the program marks it as circular and removes one of the two similar sequences as part of the error correction process (lines 81-82). However, as the authors note, this approach could incorrectly mark a linear virus genome with terminal repeats as circular (lines 310-314), and one terminal repeat will be incorrectly removed this way.

Virseqimprover checks the uniformity of the read [depth] and uses it as an indicator for [chimeric regions] in the contig (lines 83-84). While this is an interesting idea and might work with DNA virus genomes, I can imagine that it may be problematic when applied to RNA sequencing data for RNA virus genomes. RNA sequencing data often shows significant variation in site-wise depths due to variation in the transcription frequency across genomic regions. This method may also have limitations with tiled amplicon enrichment sequencing data, which, similar to RNA sequencing data, typically show non-uniform sequencing depths. Given the main thesis of the work, in addition to sequencing depth, I expect the authors to also use paired-end information, haplotype block data, read mapping quality, and the distribution of read ends to detect potential misassembled regions, or better yet, separately assemble genomes of individual virus variants co-existing within one sample.

During the extension step, Virseqimprover only chooses the longest run of “non-suspicious” region to extend (lines 89-90). However, I can imagine that, for virus genomes containing multiple repetitive elements like herpes viruses, there may be several long stretches of “non-suspicious” regions separated by “suspicious” regions corresponding to repetitive elements. In this case, focusing solely on the longest “non-suspicious” run seems unreasonable. Again, paired-end information, haplotype block data, read mapping quality, and the distribution of read ends could be used to more precisely and accurately identify (in)correctly assembled regions to work on.

Next, for each contig, “all” the reads are mapped to the edges of the contig to extend it (line 94). This approach maximises contig extension but does not account for the possibility that reads can map to multiple contigs. If reads are used to extend one contig, then they should not be used to extend other contigs if they map to them as well. Actually, with this possibility, it is clear that all contigs derived from the same sequenced sample should be considered simultaneously, not one at a time, to determine the most appropriate contig for each read based on, for example, mapping scores, or other relevant criteria.

Many of the genome annotation and visualisation tools implemented in Virseqimprover (lines 115-118) are designed for prokaryotic genome analysis, not viral genome analysis.

Finally, one of the key ideas of this program is to detect regions with non-uniform sequencing depths as “suspicious” regions. However, the program itself extends contigs without considering this point, and ends up generating contigs with non-uniform sequencing depths anyway (as shown in Figure 3(3), 5(2), and 7(3), and clearly stated in lines 241-244). This indicates yet another fundamental flaw in the program's design

Validity of the findings

There are several issues with the program’s performance evaluation and demonstration.

The authors chose only a few long contigs to demonstrate Virseqimprover (lines 120-165). However, in practice, we want to improve not only a few long contigs, but all (viral) contigs, including short ones, which can actually be more challenging to improve (and why they are highly fragmented in the first place). The authors should systematically evaluate their program using all contigs, not just a few long ones.

The authors stated that SRX7079549 was chosen here because corresponding long-read data was available for validating the results (lines 160-162). However, they instead evaluated their assembly against some assembled marine virus genome sequences, which may or may not be full-length (or at least this is unclear from the text). By comparing S7’ against AFVG 25M409, they found that the extended contig covered 98% of the AFVG 25M409 viral genome, and concluded that this result “shows that Virseqimprover successfully extends the original fragmented contig to nearly complete genome with high accuracy”. This conclusion is unfortunately logically flawed, as it is unclear if AFVG 25M409 is truly a complete genome (or at least this is unclear from the text). There seem to be fundamental flaws in the analysis and discussion of the results, or at least in how the results are described.

Continuing on this topic, instead of using long-read data or publicly available virus genomes to validate the results, it would actually be more logical to use simulated data, where the actual ground truth is completely known, making the validation more meaningful. Sequencing depth could also be completely controlled with this approach. Actually, the issue of fragmented assembly is most problematic when sequencing depth is low, which is often the case for real-world metagenomic data. The authors should test the program's limits regarding this challenge.

Given that there are currently many programs developed for extending contigs, filling gaps of draft genomes, and improving the quality of draft assemblies (lines 37-46), they should benchmark Virseqimprover against multiple programs, and not just ContigExtender (lines 166-173).

In addition, there are several strange results, or at least they are described and discussed unclearly. For example:

Lines 181-184 & Figure 3(1): The depth profile in blue on the right-hand side (around 150,000-175,000) looks suspiciously similar to the one in orange on the left-hand side (from 1 – 25,000), and drastically different from the original orange depth profile. The authors simply say that “some missing part on the left was due to the trimming of the circular region”, but the actual result seems more complicated.

Lines 185-191 & Figure3(2): The text says that the program worked on the region of length 133,375 bp from base 1 to base 133,375 and managed to extend the contig on its right to 152,362 bases. However, in Figure 3(2), the depth profile at the beginning of the improved contig is missing. Actually, the original contig length (155,659 bases) is very similar to the improved one (152,362 bases). Combined, this suggests that it might just be a matter of contig rotation rather than correction.

Lines 195-197: Given that the authors considered the region from base 133,376 to 134,543 in the original S1 contig to be a non-uniform coverage region or suspicious region (lines 185-186), seeing Figure 3(3), I don’t think the authors can say that “[the] extended contig (S2’) [has] a uniform depth of coverage along the sequence”. The authors appear to be inconsistent in their discussion.

Lastly, the test computer environment, run time, and resources required to run the program are not reported.

Additional comments

In this manuscript, Song et al. present Virseqimprover: a pipeline for improving viral contigs assembled from metagenomic sequence data. The work introduces some innovative and interesting ideas, such as using sequencing depth information to detect potentially misassembled regions and trimming sequences at each end before sequence extension. However, there are serious flaws in the work. The result presentation and writing are also rather poor. All of these combined unfortunately lead me to reject this manuscript for publication.

---

## Round 0.2 · Minor Revisions

We will be happy to accept your manuscript for publication after you have considered some of the suggestions for improvement provided by the reviewers of your current draft. Both reviewer 1 and 2 were happy with this version of the manuscript. Reviewer 3 provides a significant number of comments which should be considered as you prepare a final draft. While major changes to your paper are not required, many of the suggestions provided may help to improve the final submission.

·

Basic reporting

No comment

Experimental design

No comment

Validity of the findings

No comment

Additional comments

I appreciate the fact that the authors have made great efforts to improve the considerable flaws of the original manuscript.
Those include the ones affecting the ordering of the references in the text, the poor quality of most figures and its confusing arrangement, which have been formally improved and redistributed in a more logical way; overlapping lines have been split into separated panels and the addition of new figures can help the reader understand better the results.
-Another weakness of the submitted manuscript, the lack of details in some sections, as for example in the introduction, has been compensated by the inclusion of requested information.
-Besides, the documentation in GitHub and web server has been developed in order to correct its initial scarcity of details.
-The authors have also explicitly admitted that annotation and visualization are not strengths of their tool and it is not intended for those purposes, which is important for other users to take into consideration when opting for this tool.
-The inclusion of information regarding the performance metrics of this software is not as detailed as it could be, but I understand that probably because a generalisation of those items is very hard to achieve and thus only a general guide or pattern can be provided. For me, although not optimum, it is acceptable.
-The recurrent issue with the circularity and linear genomes has been addressed and even if the given explanation could be more convincing, it can be considered as sufficient.
-The inclusion of a simulation experiment, as suggested, to evaluate the capability of correction and extension of Virseqimprover is opportune.
-Finally, regarding the incorporation of an additional author, I do not find any substantial objection as long as it has been properly justified.
- Other more specific considerations regarding e.g. the programming would exceed my area of competence and therefore I will not comment them.

Reviewer 2 ·

Basic reporting

The text and figures have been generally improved and I have no further concerns regarding the quality of the write-up. I appreciate having my prior concerns addressed.

Experimental design

My concerns were addressed, and I have no additional concerns.

Validity of the findings

My concerns were addressed, and I have no additional concerns.

Reviewer 3 ·

Basic reporting

Throughout the manuscript: the manuscript still contains numerous incorrect uses of tense.
Lines 105-107: “Then the final extended contig is annotated for its protein content. The output contains the extended contig along with the protein annotation.” > A genome sequence cannot possibly contain protein sequences. Please revise to "genes" or "protein-coding regions," or a similar term throughout the manuscript.
Line 110: It may be clearer to separate “circularity checking” and “suspicious region detection” into two sections rather than grouping them together under "Error correction". In addition, I suggest using more precise neutral technical terms like “terminal repeat sequence detection” and “depth non-uniformity detection” to describe these processes (see my comments in the Experimental design section for further explanation / discussion).
Line 164: “reads mapped to the genome in BAM format” > This should be corrected to “reads mapped to the contig in BAM format”
Lines 179-181: “To generate the input contigs for Virseqimprover, we ran three assembly programs including FVE-novel (Tithi et al., 2023), metaSPAdes (Nurk et al., 2017), and MEGAHIT (Li et al., 2015).” > on what datasets?
Lines 185-192: This relates to my previous comment (R3-3). I’m sorry for not being clearer before, but my concern was more about self-plagiarism, not the differences between Virseqimprover and FVE-novel. Specifically, in PMID 36607772, section 3.1 says:
“The ocean virome sample SRX2912986 (labeled as Station 70) contains 18,471,506 paired-end reads with read length 151 bps. Using the GOV database containing 24,411 contigs as reference and the reads from Station 70 as input, FVE-novel generated 268 scaffolds.”
This is identical to the result described in the “Contigs generated by FVE-novel” section of this manuscript:
“In this experiment, we took the GOV database containing 24,411 contigs as the reference "genomes" and applied FVE-novel (Tithi et al., 2023) to an ocean metagenomic sample (NCBI (Sayers et al., 2022) accession number SRX2912986 (Aylward et al., 2017)) to generate viral contigs (Tithi et al., 2018). The sample contained 18,471,506 paired-end reads with an average read length 151 bp. Through FVE-novel, we produced 268 contigs, and applied Virseqimprover to the longest five contigs (hereafter labeled as S0, S1, S2, S3, and S4) to see whether the contigs could be either further extended and/or corrected for any error. Among the five contigs, S0, S1, and S2 were highly similar to each other whereas S3 and S4 were not”
Again, my point is that the authors should be clearer what was previously done, and what is new in this work. I don’t have any problem if they are using an assembly from their previous work, and if so, it must be properly acknowledged, and don’t say that “we …. applied FVE-novel (Tithi et al., 2023) to an ocean metagenomic sample .. Through FVE-novel, we produced 268 contigs…”, which is misleading. Otherwise, if the analysis was repeated independently in this work, the authors should still be clear that this was done before, and clarify if the results were the same, and if not, how do they differ (e.g. what happened to the contigs S4 and S5 reported in PMID 36607772? … see my next comment), for example.
Lines 189-190: Related to this matter, based on the identical contig lengths (Table 1 in this manuscript vs Table 1 in the PMID 36607772 paper), I believe I can deduce the contig matching pairs between the two papers (this manuscript/PMID 36607772: S0/S0, S1/S1, S2/S6, S3/S3, S4/S2). From this, I can see that contigs S4 (93,939 nt) and S5 (86,648 nt) from PMID 36607772 are actually longer than the S2 contig in this manuscript (80,620 nt), but were not analysed here. Perhaps I might have missed something, but this makes the statement “[we] applied Virseqimprover to the longest five contigs (hereafter labeled as S0, S1, S2, S3, and S4)” (lines 189-190) seems like a false statement, if not indicating data cherry picking, which is, again, highly problematic if true… Please clarify in the manuscript more precisely what was done both in PMID 36607772 and in this work.
Lines 418-421: Please correct the sentence.

Experimental design

Lines 111-115: I understand that “checking circularity is important for avoiding infinite loops during the iterative assembly process”, and I can see that the authors now say that “Virseqimprover is also applicable to linear virus genomes…” (lines 135-140). However, the text still starts by saying that circularity is used to indicate assembly completeness (lines 111-112). Given that many viruses have linear genomes (see my previous R3-7 comment), I do not see how this generally applies to viruses. In response to my previous comment, the authors replied that:
“[C]ircularity is only one of the three indicators of COMPLETE GENOME RECOVERY. To be more specific, the extension completion indicator is either assembling a circular part (i.e., the newly extended part aligns with the beginning part, meaning repetition), the number of local assembly iterations reaches the maximum threshold (i.e., 100), or after trimming 2,000 bps the sequence is still unable to be extended longer.”
I find this response unsatisfactory
Program termination criteria, which are technical, and criteria for genome completeness, which should be biological, are not the same thing. The authors shouldn’t equate the two. Also, the program can terminate simply due to a lack of sequences, generating an incomplete sequence, but this would be considered a “complete” genome by the authors’ criteria – this is false, and showcases the problem. If the authors want to claim that their program can produce complete genomes, additional biological criteria are needed. In fact, different criteria may even be needed for circular, linear, and segment genomes, or even for each specific virus group.
Additionally, many viruses contain repetitive genetic elements in their genomes, and if the repetitive regions are longer than the read lengths, then a typical assembler will terminate at these points, resulting in fragmented contigs with repeating sequences at the ends. Furthermore, some linear virus genomes have terminal repeats. According to the algorithm implemented in Virseqimprover, all contigs such as these would be flagged as circular, and one of the terminal repeats would be removed, which is incorrect in principle.
I have a suggestion. Since the presence of terminal repeats on a contig can mean many things, I suggest using a neutral term like “containing terminal repeats” instead of jumping to the conclusion that a contig with terminal repeats represents a complete genome assembly. The authors can then explain from the technical perspective why they decide to do this (i.e. to avoid the infinite loop, etc.), and then discuss how to interpret the results within the context of these uncertainties.
Lines 111-115 and figure 1: My understanding is that if Virseqimprover detects terminal repeats in a contig, it skips the depth non-uniformity (chimera) detection step. I still don’t understand why this is the case. I think the program should check every contig for chimeric sequences, regardless of whether terminal repeats are present. Please justify this design choice; otherwise, consider modifying the program so that depth non-uniformity is checked for all input contigs.
Lines 116-123 and figure 2: It doesn’t seem like Virseqimprover checks the orientation of the terminal repeats during the trimming process. If this is the case, it should.
Lines 122-123: Since terminal repeats can be a genuine feature in the genomes of some viruses, the authors should offer users an option to decide for themselves whether to trim the terminal repeats or not. It would be very helpful if the program flags the detected terminal repeats in the result report to help users do this.
Lines 132-133: The statement “Virseqimprover chooses the longest true non-suspicious region to extend during the extension step” raises concerns. My previous comment regarding this was that “focusing solely on the longest “non-suspicious” run seems unreasonable” since “there may be several long stretches of “non-suspicious” regions [scattering across a contig]” (R3-10). The authors replied that “[s]ince Virseqimprover is not a gap filler, choosing the longest “non-suspicious” region and start correcting and extending from it is reasonable and intuitive”. Frankly, I don’t understand how this addresses my concern.
I also suggested that “paired-end information, haplotype block data, read mapping quality, and the distribution of read ends could be used to more precisely and accurately identify (in)correctly assembled regions to work on”. The authors acknowledged that this information could be useful but, “Virseqimprover can also handle single-end reads where paired-end information is not available”. This is irrelevant. Even when using single-end data, haplotype block information, read mapping quality, and read-end distribution are still attainable, right? And, how about when pair-end reads are analysed?
Line 145: The statement “…for each contig, all the reads are mapped to the edges of the contig…” still raises concerns. My previous comment regarding this was that “[i]f reads are used to extend one contig, then they should not be used to extend other contigs …” and that “all contigs derived from the same sequenced sample should be considered simultaneously, not one at a time, to determine the most appropriate contig for each read …” (R3-11). The authors responded by saying that “when considering those conserved regions, mapping to either contig can be correct if they have uniform depth of coverage since our goal is to try to correct and extend contigs”. However, I still do not understand this explanation. Furthermore, I do not see any part of the program that checks whether contigs are conserved.
Lines 172-175: the text says “[o]ther than the method, other databases like Prokaryotic Virus Orthologous Groups (pVOGs) (Grazziotin, Koonin & Kristensen, 2017), and annotation tools like DRAM (Shaffer et al., 2023) and Pharokka (Bouras et al., 2023) can also be used to annotate the final contigs.”. Please confirm if these databases and tools are actually implemented in the program. If not, they should be.
Line 439: While the authors have already acknowledged the program’s limitation with amplicon enrichment sequencing data, they still haven’t discussed this potential limitation with RNA sequencing data. RNA sequencing data often show significant variation in site-wise depth due to variation in transcription frequency across genomic regions. This raises a possibility that an assembled RNA virus genome might be falsely flagged as chimeric by the program. Please address this issue.

Validity of the findings

Reply to R3-14: The author replied that “[o]ur main goal was not to extend every contig in the metagenomic samples where the amount can be huge. Instead, the main goal was to correct and extend a small amount of contigs that researchers are particularly interested in.”. This is actually a good point of discussion about the scope of the program. Please include this in the manuscript.
Lines 251-359: In addition to the depth information and BLAST results, I think it is important to show and analyse the raw read-mapping alignments both before and after applying Virseqimprover. By examining pair-end locations, read-mapping direction, and alternative allele frequencies, the authors should discuss the differences between them to justify whether the extended contigs were really improved and correctly extended. The read mapping should be performed across multiple contigs at the same time (e.g., S0-S4 and S5-S7) to see whether the same reads were used to extend multiple contigs or not, and if so (which I believe might show up as secondary alignments), discuss what this means for the interpretation of the results.
This analysis is particularly important for contig S3, where Virseqimprover and Geneious gave different results (lines 307-323) – when two methods give different results for the same dataset, it typically indicates that one may be wrong (if not both). In this case, looking at the raw read-mapping alignments could help determine which result is correct. If the authors believe both results are correct despite the differences (as seems to be suggested by the current text ?), they should provide a more thorough discussion of the discrepancy and its implications for interpreting the results.
Lines 325-330: How does S4’ compare to the one obtained from Geneious (line 210)?
Lines 336-339: Discuss possible reasons for why the program was unable to generate the full-length genome, and illustrate the results in the figure, similar to Figure 7B.

Additional comments

Some of my previous comments have been addressed, but several issues remain regarding program design, data interpretation, and result validation. I believe the manuscript still need significant revisions before it can be considered for publication.

---

## Round 0.3 · accepted · Accept

Thank-you for submission of a revised version of your manuscript that incorporated your response to the remaining reviewer's critiques. We are now happy to accept it for publication.